# Establishment of an Integrated CRISPR/Cas9 Plasmid System for Simple and Efficient Genome Editing in Medaka In Vitro and In Vivo

**DOI:** 10.3390/biology12020336

**Published:** 2023-02-20

**Authors:** Zeming Zhang, Jie Wang, Jianeng Li, Xiang Liu, Lei Liu, Changle Zhao, Wenjing Tao, Deshou Wang, Jing Wei

**Affiliations:** 1Key Laboratory of Freshwater Fish Reproduction and Development (Ministry of Education), Integrative Science Center of Germplasm Creation in Western China (CHONGQING) Science City, Laboratory of Aquatic Science of Chongqing, School of Life Sciences, Southwest University, Chongqing 400715, China; 2Sichuan Province Yuechi Middle School, Guang’an 638300, China

**Keywords:** CRISPR/Cas9, knock-out, knock-in, fish cultured cells, U6 promoter, medaka

## Abstract

**Simple Summary:**

The application of CRISPR/Cas9 genome editing in fish is limited to date. An easy-to-use, economical and effective genome editing system is greatly needed. In this study, we established an integrated pCas9-U6sgRNA plasmid system, the sgRNA of which was driven by U6 promoters from different fish species. Taking medaka as an example, our results suggest that pCas9-U6sgRNA driven by endogenous U6 promoter (pCas9-mU6sgRNA) can very effectively mediate gene knock-out in medaka cultured cells, but not by exogenous U6 promoter. The gene editing efficiency of pCas9-mU6sgRNA was up to 93.7% with no detectable off-target. By one-cell embryo microinjection, pCas9-mU6sgRNA can effectively mediate gene knock-out in vivo as well. Furthermore, the gene knock-in at a specific site was efficiently achieved in vitro as well as in vivo through application of pCas9-mU6sgRNA. In conclusion, we have successfully developed a simple, low-cost and effective CRISPR/Cas9 system suitable for medaka gene knock-out and knock-in at a specific site in vitro and in vivo. This study provides an insight into other fish gene manipulation and greatly promotes functional gene studies.

**Abstract:**

Although CRISPR/Cas9 has been used in gene manipulation of several fish species in vivo, its application in fish cultured cells is still challenged and limited. In this study, we established an integrated CRISPR/Cas9 plasmid system and evaluated its efficiency of gene knock-out or knock-in at a specific site in medaka (Oryzias latipes) in vitro and in vivo. By using the enhanced green fluorescent protein reporter plasmid pGNtsf1, we demonstrate that pCas9-U6sgRNA driven by endogenous U6 promoter (pCas9-mU6sgRNA) mediated very high gene editing efficiency in medaka cultured cells, but not by exogenous U6 promoters. After optimizing the conditions, the gene editing efficiencies of eight sites targeting for four endogenous genes were calculated, and the highest was up to 94% with no detectable off-target. By one-cell embryo microinjection, pCas9-mU6sgRNA also mediated efficient gene knock-out in vivo. Furthermore, pCas9-mU6sgRNA efficiently mediated gene knock-in at a specific site in medaka cultured cells as well as embryos. Collectively, our study demonstrates that the genetic relationship of U6 promoter is critical to gene editing efficiency in medaka cultured cells, and a simple and efficient system for medaka genome editing in vitro and in vivo has been established. This study provides an insight into other fish genome editing and promotes gene functional analysis.

## 1. Introduction

A powerful genome editing tool, clustered regularly interspaced short palindromic repeat (CRISPR)-CRISPR associated protein 9 (Cas9), has been successfully applied to a variety of species in vivo as well as in vitro, pushing the research of gene functions into an unprecedented new era [1,2,3,4]. Cultured cells including cell lines derived from various species are useful models in the study of virology [5], environmental toxicology [6,7], genetic breeding [8], resources conservation [9], molecular mechanisms and so on. Compared with whole organism, cultured cells cannot only save cost and time, but also minimize the influence of genetic heterogeneity and other influencing factors, such as neuroendocrine factors, interactions with other cells and ethical concerns [10,11,12]. In mammals, a variety of gene knock-out and knock-in cell lines have been established. Lindner et al. report that a genome-wide gene knock-out model based on CRISPR/Cas9 was established, which greatly promotes high throughput identification of potential factors affecting cell biological processes [13]. These studies are undoubtedly of great significance for in vivo cell tracking, gene functional analysis and so on.

Although the CRISPR/Cas9 system has been successfully applied to genome editing in some fish species, such as zebrafish (*Danio rerio*) [14], medaka (*Oryzias latipes*) [15], Nile tilapia (*Oreochromis niloticus*) [16], southern catfish (*Silurus meridionalis*) [17], rare minnow (*Gobiocypris rarus*) [18] and Atlantic salmon (*Salmo salar*) [19,20,21], its application in fish cultured cells is still challenged and limited [22,23,24,25,26,27,28,29]. In addition to the advantages of cultured cells compared with whole organism, genome editing in cultured cells are not affected by fish spawning cycles. For example, some fishes lay eggs once a year [30], which seriously hinders the research of their gene functions. Therefore, an efficient genome editing tool in fish cultured cells is urgently needed. Multiple attempts were made, such as synthetic RNA [28], ribonucleoprotein (RNP) [22,27,29], plasmid [24,29] or lentivirus [25]-mediated expression systems for Cas9/sgRNA genome editing, but only a few of them achieved gene mutational cell lines. Liu et al. report that efficient gene editing was achieved through electroporation of medaka cultured cells with pre-formed Cas9/sgRNA RNP complex, but failed through transfecting with synthetic sgRNA, plasmid or lentivirus mediated expression systems of Cas9 and sgRNA driven by mammalian U6 promoter [22]. Nevertheless, Hamar et al. recently report that genomic mutations in a tilapia (*Oreochromis mossambicus*) brain cell line failed to obtain through transfecting of cells with Cas9/sgRNA RNP complex using Invitrogen Lipofectamine CRISPRMAX transfection reagent system, whereas effective genome mutations were achieved through obtaining cells with Cas9 constitutive expression driven by an endogenous tilapia elongation factor 1α promoter and then transfected with sgRNA expression vector driven by tilapia U6 promoter [26]. These studies imply that the species relationship between U6 promoters and the transfected fish cultured cells might be a determinant factor affecting gene editing efficiency. U6 snRNA (U6) is one of the six abundant and capped small nuclear RNAs which is involved in the splicing of pre-mRNAs and transcribed by RNA polymerase III [31,32]. U6 promoter is generally used to promote expression of shRNA or sgRNA in vector-based RNA interference or genome editing [33,34]. Moreover, the activity of U6 promoter shows species-specificity behavior to some extent [35]. Nevertheless, the correlation between the genetic relationship of U6 promoters from different fish species and genome editing efficiency in vector-based CRISPR/Cas9 systems is still elusive, and remains to be investigated. Moreover, these above methods are high cost, have specific equipment requirements, high technical requirements and a high risk of off-target [26]. Therefore, an easy-to-use, efficient and economical genome editing tool is greatly needed in fish cultured cells.

In this study, using an enhanced green fluorescent protein (EGFP) reporter plasmid, the gene editing efficiency of an integrated CRISPR/Cas9 plasmid system, sgRNA of which is driven by U6 promoters derived from medaka, Nile tilapia or zebrafish, was evaluated in medaka cultured cells, respectively. Based on these, an integrated CRISPR/Cas9 plasmid system which mediated very high gene knock-out or knock-in efficiency in medaka in vitro as well as in vivo was successfully developed. This study will provide an insight into fish genome manipulation and greatly promote functional gene studies.

## 2. Materials and Methods

### 2.1. Fish

Medakas were kept in re-circulating, aerated water tanks at 26 °C [36]. All animal experiments were carried out in accordance with the regulations of the Guide for Care and Use of Laboratory Animals and approved by the Institutional Animal Care and Use Committee of Southwest University (NO. IACUC-20181015-12).

### 2.2. Plasmids

The genome editing vector pCas9-U6sgRNA was constructed with the backbone of pcl2EGFP-pzU6sgRNA-pCVzCas9. Original cardiac myosin light chain 2 promoter and EGFP gene were replaced by the simian virus 40 enhancer and early promoter (SV40) and neomycin resistance gene (NeoR) for G418 selection, named pCas9-U6sgRNA. For further optimization, zebrafish U6 promoter (zU6) was replaced by medaka U6 promoter (mU6) or Nile tilapia U6 promoter (NtU6), respectively. Cas9 is zebrafish codon-optimized version of *Streptococcus pyogenes* Cas9. Different sgRNA was introduced by using pEASY^®^-Basic Seamless Cloning and Assembly Kit (TransGen Biotech, Beijing, China). Briefly, take Nile tilapia validated *sf1* sgRNA (5′-GGCTGTGTACTGGTACTGGG-3′) [37] insertion in pCas9-zU6sgRNA, for example. Primer pairs of U6Mul-F (5′-CTTGACGAGTTCTTCTGAACGCGTCTCGAGCCTCTAGA-3′) and zU6sgNtsf1-R1 (5′-CCCAGTACCAGTACACAGCCCGAACCAAGAGCTGGAGGGA-3′) were used to amplify the upstream sequence, primer pairs of sgNtsf1-F1 (5′-GGCTGTGTACTGGTACTGGGGTTTTAGAGCTAGAAATAGC-3′) and scaffoldSal-R (5′-AATTGGCGGTCGACTGGCGTAATAGCCAAC-3′) were used to amplify the downstream sequence. The pCas9-zU6sgRNA plasmids were linearized by restriction enzyme *Mlu* I and *Sal* I. These three kinds of DNA fragments were assembled together according to manufacturer’s instructions.

Reporter plasmid pGNtsf1 was constructed with the backbone of pCVpf [38,39] which has been reported to be used in medaka cultured cells. It was constructed by inserting in frame the puromycin acetyltransferase gene to *egfp* in pEGFP-N1 (Clontech, Mountain View, CA, USA). The sgRNA target sequence of Nile tilapia *sf1* gene with 5′-TGG-3′ PAM sequence [37] was inserted into EGFP gene coding sequence in pCVpf. At the same time, a 20-bp (base pair) length nucleic acid sequence (5′-CGAGGGCGAGGGCGATGCCA-3′) corresponding with the upper part of the inserted cite was introduced behind in order to mediate homologous recombination repair when pGNtsf1 was cut off (Figure 1B).

Donor plasmid used in gene knock-in was constructed with the backbone of pCVpf with the removal of CMV promoter. The 1232-bp length left homologous arm and 1249-bp length right homologous arm to the sgRNA target sequence of medaka *actb2* (*actin*, *beta 2*) were inserted into each side of the promoter-less puromycin acetyltransferase-EGFP fusion gene, respectively.

### 2.3. Cell Culture and Cell Transfection

Medaka spermatogonial stem cell line SG3 and embryonic stem cell line MES1were maintained in ESM4 at 28 °C as described [40,41]. DNA transfection was performed using TransIT-X2^®^ Dynamic Delivery System (Mirus Bio, Madison, WI, USA) according to manufacturer’s instructions. At 24–72 h after transfection, photographing or genomic DNA extraction were performed. As for G418 selection, cells were cultured for another 7 days with culture medium containing 1000 μg/mL G418.

### 2.4. Microinjection

Medaka one-cell embryos were microinjected with 50–150 pg plasmids as previously described [42]. EGFP signals were observed at 24 h after injection, and then genomic DNAs were extracted for PCR (polymerase chain reaction) amplification.

### 2.5. HMA

Heteroduplex mobility assay (HMA) was used to monitor the cleavage efficacy of pCas9-U6sgRNA as previously described [43], which has been shown to be an efficient method to detect and screen small gene sequence alterations in the genome. Briefly, genomic DNAs were extracted from cells or embryos by phenol-chloroform method after 24 h digestion with STNE buffer (20 mM Tris-HCl, 5 mM EDTA, 0.4M NaCl, 10g/L SDS, 100 mg/mL proteinase K, and 200 ng of genomic DNA was used for PCR amplification. PCR was run using primer sets specific for target genes (Appendix A) for 35 cycles (95 °C for 30 s, 55 °C for 30 s and 72 °C for 30 s) with 2 × Taq Master Mix (Vazyme, Nanjing, China). PCR amplification products were denatured at 94 °C for 3 min and slowly cooled down to room temperature to form heteroduplex or homoduplex. Subsequently, 5 µL of PCR products were separated on 12% polyacrylamide gels electrophoresis (PAGE) in TBE buffer using a PowerPac basic electrophoresis unit (Bio-Rad, Hercules, CA, USA). Gels were photographed by a bioimaging system Fusion FX7 (Vilber Lourmat, Paris, France).

### 2.6. TIDE Analysis

TIDE (Tracking of Indels by Decomposition) webtool version 3.3.0 (http://tide.nki.nl, accessed on 15 May 2022) was used to quantitate INDEL mutation frequency [26,44]. Briefly, PCR amplicons of control and pCas9-U6sgRNA treated cell DNA were purified and sent to the Beijing Genomics Institute (BGI) company for Sanger sequencing. Resulting chromatogram sequence files were uploaded to the TIDE website and analyzed using default settings except that the INDEL size range was set to  +  or −50 bp.

### 2.7. Statistical Analysis

All data were processed with GraphPad Prism 8.0.1 (GraphPad Software, Inc., San Diego, CA, USA). Data from at least three distinct experiments are provided as mean ± SD. Independent sample t-tests were employed to discover significant differences in mean values between the two groups. The mean values of multiple groups were compared using one-way ANOVA (and nonparametric or mixed) tests. For all tests, the statistical significance criteria were set at *p*  <  0.05.

## 3. Results

### 3.1. Construction and Evaluation of an Integrated Plasmid-Based CRISPR/Cas9 System in Medaka Cultured Cells

An integrated plasmid-based CRISPR/Cas9 system was constructed (Figure 1A). The expression of Cas9 was driven by CMV promoter. The expression of sgRNAs was driven by U6 promoter derived from medaka, Nile tilapia or zebrafish, thereby named as pCas9-mU6sgRNA, pCas9-NtU6sgRNA and pCas9-zU6sgRNA, respectively. The CRISPR/Cas9 system contains a neomycin resistance gene driven by SV40 promoter to enrich transfected cells.

The EGFP reporter plasmid pGNtsf1 containing a validated target sequence of Nile tilapia *sf1* [37] was constructed and used to intuitively monitor the gene editing efficacy of pCas9-U6sgRNA in medaka cultured cells. The SG3 cells were co-transfected with pGNtsf1 and pCas9-U6sgRNA targeting Nile tilapia *sf1* (named as pCas9-U6sgNtsf1). When pGNtsf1 was cut off at the inserted target sequence of Nile tilapia *sf1* gene, the upper sequence and introduced lower sequence of this inserted target sequence will work as homologous arms, and the correct reading frame of *egfp* will be restored when homologous recombination repair occurs, and then EGFP signals can be observed. At 48 h after co-transfecting SG3 with pCas9-mU6sgNtsf1 and pGNtsf1, over 10% of EGFP positive (EGFP^+^) cells were observed (Figure 1C). After co-transfecting cells with pCas9-NtU6Ntsf1 and pGNtsf1, the rate of EGFP^+^ cells were much lower (Figure 1D), and the rate of EGFP^+^ cells was rare after transfecting cells with pCas9-zU6sgNtsf1 and pGNtsf1(Figure 1E). No EGFP signals were observed after the cells were transfected only with pGNtsf1 (Figure 1F). It is worth mentioning that there was transfection efficiency over 40% in SG3 using the positive control plasmids in our previous studies [45,46]. In this study, due to the limited efficiency of homologous recombination and repair, the actual editing efficiency may be much higher than the efficiency directly observed. Meanwhile, similar results were achieved in MES1 cells [47] (Appendix A). Our data unambiguously demonstrate that the genetic relationship between the U6 promoters in CRISPR/Cas9 co-expression system and medaka cultured cells is a decisive factor affecting the efficiency of gene editing.

### 3.2. Dosage Optimization of pCas9-U6sgRNA in Medaka Cultured Cells

To maximize gene editing efficiency, different dosages (10, 50, 100, 200, 300, and 400 ng) of pCas9-mU6sgNtsf1 together with 200 ng reporter plasmids pGNtsf1 were co-transfected into SG3 cells in a 24-well plate, respectively. With the increase in pCas9-mU6sgNtsf1 dosage, the percentage of EGFP^+^ cells increased and peaked at the dosage of 300 ng. (Figure 2). Hence, this dosage (300 ng) was used at the subsequent experiments. By the way, a few fluorescence signals were also observed even if SG3 cells were transfected with pCas9-mU6sgNtsf1 at a very low dosage (10 ng) (Figure 2A), reflecting the high gene editing efficiency of pCas9-mU6sgRNA in medaka cultured cells.

### 3.3. Endogenous Gene Knock-Out Mediated by pCas9-mU6sgRNA in Medaka Cultured Cells

In order to further evaluate the gene editing effect of pCas9-mU6sgRNAs on endogenous genes in the genome, eight sgRNAs targeting four genes (two sgRNAs every one gene) were designed, four of which targeted *ptch1* (*patched 1*) and *ptch2* (*patched 2*), the other four validated sgRNAs targeted *tmem104* (*transmembrane protein 104*) and *sytl5* (*synaptotagmin-like 5*) [20] (Appendix A), respectively. After SG3 cells were transfected with the eight pCas9-mU6sgRNAs, respectively, and then enriched by G418 selection, gene sequence alterations were evaluated by HMA and TIDE analysis.

Multiple heteroduplex or homoduplex DNA bands were observed in all of the eight pCas9-mU6sgRNAs, whereas no heteroduplex or homoduplex DNA bands were observed in the control of each group (Figure 3A–D). To quantitate INDEL mutation frequency, the PCR products of each sample were subjected to TIDE analyses. The indel rates of the targets ranged from 31.2% to 93.7%, except for the relatively low indel rates from pCas9-mU6sgRNA targeting *ptch2* sgRNA1 (14.8%) and *sytl5* sgRNA1 (6.5%), respectively (Figure 3E). To further confirm the indels mediated by pCas9-mU6sgRNA, the PCR products were recovered and subcloned for DNA sequencing. Five clones of each sgRNA were sequenced and at least one type of indel by each pCas9-mU6sgRNA was obtained (Appendix A).

To evaluate the risk of off-target of pCas9-mU6sgRNA in medaka cultured cells, potential off-target sites of three sgRNAs were predicted by CCTop (CRISPR/Cas9 target online predictor) (https://crispr.cos.uni-heidelberg.de, accessed on 15 October 2022). There are four potential off-target sites, i.e., one for *ptch1* sgRNA1, one for *ptch1* sgRNA2 and two for *tmem104* sgRNA1 (Appendix A). The corresponding samples were analyzed by HMA. No additional heteroduplex or homoduplex DNA bands were observed in the four predicted sites compared with the control, indicating that there were no detectable mutations existed at the potential off-target sites using the pCas9-mU6sgRNAs in SG3 cells (Figure 3F).

### 3.4. Mutational Single Cell Clones Generated with pCas9-mU6sgRNA

The acquisition of monoclonal mutant cell lines is an important step in the study of gene function in vitro, which is not only related to the function of gene of interest, but also closely related to the growth rate and screening strategy themselves. To generate cell clones with mutations by using pCas9-mU6sgRNA, SG3 cells were transfected with pCas9-mU6sgRNA targeting *sall4* (*spalt*-like transcription factor 4), named as pCas9-mU6sgsall4 hereafter, and then enriched by G418 selection for 7 days. Through dilution and subculture, single-cell clones were obtained followed by genomic DNA extraction.

Genomic DNA PCR amplification products were analyzed by HMA (Figure 4A) and subcloned into plasmids for sequencing. Three single-cell clones possessing different mutations were identified (Figure 4B). Clone 1 contains a 1-bp insertion, clone 2 contain a 1-bp insertion and 9-bp deletion in one allele, and 3-bp insertion and 28-bp deletion in the other allele, and clone 3 contains a 219-bp insertion (Figure 4B). Our study suggests that mutated cell clones can be readily generated from medaka cultured cells by pCas9-mU6sgRNA.

### 3.5. Gene Knock-Out Mediated by pCas9-mU6sgRNA In Vivo

To test the gene editing efficiency of pCas9-mU6sgRNA in vivo, pCas9-mU6sgNtsf1 (75 pg) and the reporter plasmid pGNtsf1 (75 pg) were co-injected into one-cell embryos of medaka. Embryos injected only with pGNtsf1 were set as the control.

EGFP signals were obviously observed in animal pole regions of the embryos at 24 h after co-injection with pCas9-mU6sgNtsf1 and pGNtsf1 (Figure 5A–A″), whereas no EGFP signals were observed in the control embryos (Figure 5B–B″). Meanwhile, pCas9-mU6sgptch1 targeting endogenous *ptch1* was injected into one-cell embryos of medaka, genomic DNAs of each embryo were extracted at 24 h after injection and analyzed by HMA. Distinct heteroduplex or homoduplex DNA bands were observed in 5 of the 11 samples (Figure 5C). Our data suggest that pCas9-mU6sgRNA can also mediate gene knock-out in vivo.

### 3.6. Gene Knock-In Mediated by pCas9-mU6sgRNA In Vitro and In Vivo

CRISPR/Cas9 can induce DNA double strand breaks at specific sites, and foreign DNA fragments can be integrated into specific genomic sites by homology-directed repair in the presence of donor templates. To assess the gene knock-in effect of pCas9-mU6sgRNA, a sgRNA targeting medaka *actb2* was designed and introduced into pCas9-mU6sgRNA (named as pCas9-mU6sgactb2). This sgRNA target sequence of medaka *actb2* overlaps with the stop codon of *actb2* (Figure 6A).

In the presence of the donor plasmid, which contains a promoter-less puromycin resistance gene fused EGFP gene, EGFP signals might be observed when pCas9-mU6sgactb2 mediates gene knock-in and in-frame of medaka *actb2*.

At 24 h after co-transfection of SG3 cells with pCas9-mU6sgactb2 and the donor plasmid, a few EGFP^+^ cells were observed (Figure 6B–C″), whereas no EGFP^+^ cells were observed after transfection of SG3 cells only with the donor plasmid (Figure 6D–D″). When puromycin (5 μg/mL) was supplemented, EGFP negative cells were gradually eliminated, whereas EGFP^+^ cells survived (Figure 6C–C″), suggesting that the promoter-less puromycin acetyltransferase-EGFP fusion gene was integrated into SG3 genome and expressed. Accurate integration at the designed site was further confirmed using PCR and sequencing with primers as indicated (Figure 6A and Appendix A).

To further evaluate the viability of gene knock-in mediated by pCas9-U6sgRNA in vivo, the pCas9-mU6sgactb2 and the donor plasmid described above were injected into one-cell embryos of medaka. Embryos injected only with the donor plasmid were set as the control. At 24 h after co-injection, EGFP signals were observed in the animal pole region among the pCas9-mU6sgactb2 and donor plasmid co-injection embryos (n > 3) (Figure 6E–E″), whereas no EGFP signals were observed in the control embryos (Figure 6F–F″).

Collectively, our data suggest that pCas9-U6sgRNA can effectively mediate gene knock-in at a specific site in medaka in vitro as well as in vivo.

## 4. Discussion

In this study, by using the reporter plasmid pGNtsf1, our results intuitively and convincingly suggest that the genetic relationship between the U6 promoter in CRISPR/Cas9 co-expression system and medaka cultured cells is a decisive factor affecting the efficiency of genome editing. Furthermore, an integrated plasmid pCas9-U6sgRNA for genome editing in vitro as well as in vivo has been established. Cas9/sgRNA-RNP requires specific equipment, technical requirements and high-cost specific reagents [22,26,29]. Transfection of stable Cas9 expression cells with synthetic sgRNA or plasmid may increase the risk of off-target due to the continuous expression of Cas9 [26]. In addition, the sgRNA synthesized in vitro is easy to degrade [28]. In contrast, pCas9-U6sgRNA containing endogenous U6 promoter has the advantages of economy, simplicity and efficiency. Just through conventional vector construction and cell transfection, highly efficient genome editing at desired sites can be generated in medaka cultured cells, and the editing efficacy of target gene can be evaluated by direct EGFP signals observation or HMA and TIDE analysis within a few days. Additionally, clonal mutational cell lines can be obtained from the mixed transfected cell populations by G418 selection and limiting dilution analysis. The simplicity of the pCas9-U6sgRNA opens up the possibility to carry out genome-wide targeted mutagenesis in medaka cultured cells to screen the functional deletion mutation of cell phenotypes and serves as the source of sgRNA design for future research. Meanwhile, pCas9-mU6sgRNA can efficiently mediate gene knock-out and knock-in at a specific site in embryos as well. Although it may be better to gain genome editing off springs by microinjection of sgRNA and Cas9 mRNA or mixed RNPs into one-cell embryos, the pCas9-mU6sgRNA provides an alternative choice for the study of genes with lethal effects at the early stage of embryonic development. Because the expression of sgRNA and Cas9 in pCas9-mU6sgRNA may be initiated until blastula, the delayed gene editing might be conducive to study the functions of these lethal genes in embryonic development at the late stage.

Several factors are proposed to hurdle the application of CRISPR/Cas9-mediated genome editing in fish cultured cells. First, compared with mammalian cells, DNA fragments are more difficult to enter fish cells, which may be caused by the low temperatures of incubation in fish cell cultures and increased saturation of phospholipids in fish cell membranes [27,48,49]. An alternative method of plasmid delivery is to introduce sgRNA and/or Cas9 into cells using virus. Gratacap et al. report that effective genome editing was achieved using lentivirus delivery system for CRISPR/Cas9 mediated genome editing in a salmonid cell line CHSE-214 [25]. Nevertheless, the random integration of viral DNA into the host genome may cause insertional mutation in target cells, and constitutive expression of CRISPR/Cas9 elements may lead to an increased risk of off-target effect. In addition, according to our experience, the transfection efficiency of fish culture cells such as Nile tilapia cells by lentivirus was low (unpublished data). All these factors limit the application of virus vectors. Of note, there are several reports of electroporation transfection of the Cas9 protein/sgRNAs RNP in fish cultured cells including medaka fish cells SG3 [22], rainbow trout (*Oncorhynchus mykiss*) RTG-2 [27], Chinook salmon (*Oncorhynchus tshawytscha*) CHSE-214 [48] and Atlantic salmon (*Salmo salar*) ASK-1 and SHK-1 [29] with high gene editing efficiencies [26]. Second, there is lack of a well characterized and efficient plasmid-based CRRISPR/Cas9 expression system which works well in fish cultured cells. Although there are several reports within which gene editing in Cyprinidae fish cell lines including the common carp caudal fin (KF-1) cell line [50] and grass carp kidney cells CIK [23] was observed using sgRNA expression system driven by mammalian U6 promoter, the gene editing efficiency was not evaluated. Sebastian et al. developed a bicistronic vector using zebrafish U6 promoter for the expression of the sgRNA in fish cell line CHSE/F [24]. A recent study by Hamar et al. shows that high gene editing efficacy was successfully achieved through transfecting tilapia brain cells with sgRNA expression vector driven by tilapia U6 promoter [26], implying promoter of sgRNA is a critical factor affecting gene editing efficiency in fish cultured cells. Nonetheless, the species relationship between U6 promoter and transfected fish cultured cells remains elusive.

U6 promoter belongs to polymerase III promoter, it is widely used in expression of shRNA or sgRNA [34,51]. In fish, zebrafish U6 promoter has been isolated and characterized for shRNA expression in Nile tilapia cell extracts [33] and sgRNA expression in a Chinook salmon cell line [24]. In this study, to construct sgRNA expression plasmid, U6 promoters from three fish species with distant evolutionary distance, i.e., medaka, Nile tilapia and zebrafish, were isolated and characterized. By multiple sequence alignment of U6 promoters, although all of the three fish species contain the homology elements including SphI postoctamer homology elements, proximal sequence elements and TATA box, the sequence and the arrangement of the homology elements is not conserved (Appendix A). Previous studies also show that species specific transcription factors may affect the activity of the U6 promoter in the cellular environment [35]. In our preliminary study, U6 promoter from zebrafish was used to express sgRNA in medaka SG3 cells. To our disappointment, few EGFP signals could be observed after co-transfection with the reporter plasmid pGNtsf1 and the pCas9-zU6sgNtsf1, and no indels could be detected in genomic DNA of SG3 cells after transfection with pCas9-zU6sgNtsf1. Nonetheless, after SG3 was co-transfected with pGNtsf1 and the pCas9-mU6sgNtsf1, the sgRNA of which was driven by endogenous (medaka) U6 promoter, a large number of EGFP signals could be observed, the number of EGFP signals derived from SG3 cells transfected with pCas9-NtU6sgNtsf1 driven by the Nile tilapia U6 promoter was less (Figure 1C–F). In genomic DNA editing, the highest indel rate was up to 93.7% in SG3. Similar results could be observed in another medaka cell MES1 (Appendix A). Collectively, our results provide solid evidence that the genetic relationship of U6 promoter in CRISPR/Cas9 is a decisive factor affecting the efficiency of genome editing in fish cultured cells.

It is known that evaluating the gene editing efficiency of a designed sgRNA in fish cells is time-consuming and labor-intensive. If the designed sgRNA does not work or the efficiency is low, the research will become more difficult. In the present study, to intuitively and rapidly evaluate the gene editing efficiency of sgRNA in fish cultured cells, we constructed the EGFP reporter plasmid pGNtsf1. In this plasmid, the known sgRNA target sequence of Nile tilapia *sf1* plus a 20-bp EGFP coding sequence next to the 5′ end of the inserted sgRNA target sequence was inserted into the EGFP coding sequence. During the CRISPR/Cas9 mediated genome cleavage repair process, the double 20-bp sequences upstream and downstream of the sgRNA target sequence were used as homologous repair templates, resulting in the restoration and expression of the full coding sequence of EGFP. Therefore, we can directly and quickly evaluate the gene editing efficiency of sgRNA in a fish cellular context through observation of EGFP signals, which greatly saves time costs, economic costs and labor costs.

## 5. Conclusions

In our study, a method system that can intuitively and quickly evaluate the gene editing efficiency of the plasmid-based CRISPR/Cas9 system in fish cultured cells has been established, which might be applied to screen sgRNAs or evaluate the activity of U6 promoter in CRISPR/Cas9 system. The gene editing efficiency of pCas9-U6sgRNA, the sgRNA of which is driven by U6 promoter derived from medaka, Nile tilapia or zebrafish, were evaluated in medaka cultured cells and our data demonstrate that the genetic relationship between the U6 promoter in CRISPR/Cas9 co-expression system and medaka culture cells is a decisive factor affecting the efficiency of gene editing. Finally, an integrated CRISPR/Cas9 plasmid system pCas9-U6sgRNA for simple and efficient gene knock-out and knock-in in medaka in vitro and in vivo has been successfully established. This study provides new ideas and insights for fish genome editing, and will promote fish gene functional research.

## Figures and Tables

**Figure 1 biology-12-00336-f001:**
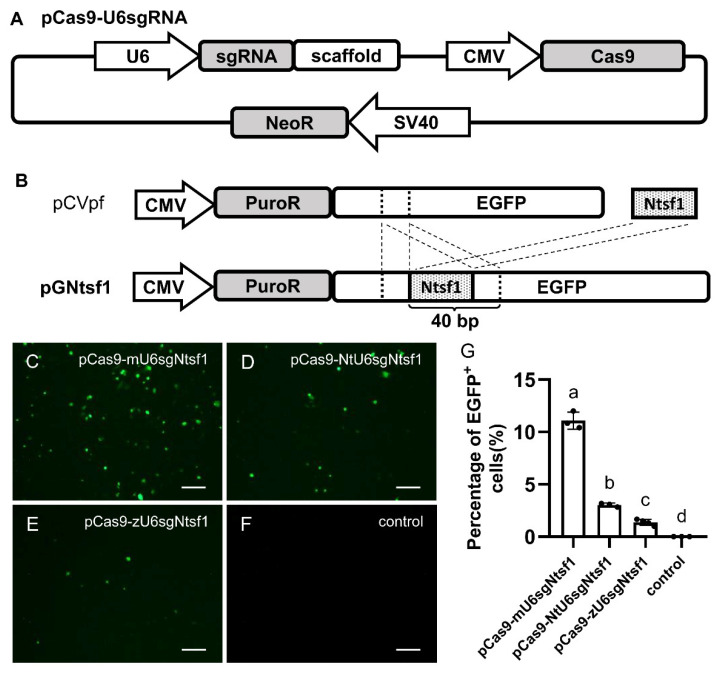
The effect of U6 promoters derived from different species on gene editing efficacy in medaka cultured cells. The integrated pCas9-U6sgRNA plasmids containing U6 promoter derived from different species was constructed. Then it was transfected into medaka cultured cells SG3 together with pGNtsf1 reporter vector. The gene editing efficacy was evaluated by observing EGFP signals 48 h after transfection. (**A**) Diagram of pCas9-U6sgRNA plasmid. (**B**) Diagram of construction strategy of the reporter plasmid pGNtsf1. pCVpf was used as backbone. (**C**–**F**) SG3 cells were co-transfected with pCas9-U6sgRNA and pGNtsf1. The sgRNA targeting Nile tilapia *sf1* in pCas9-U6sgRNA is driven by U6 promoters derived from medaka (pCas9-mU6sgNtsf1) (**C**), Nile tilapia (pCas9-NtU6sgNtsf1) (**D**) and zebrafish (pCas9-zU6sgNtsf1) (**E**), respectively. SG3 cells transfected only with pGNtsf1 were set as negative control (**F**). Scale bars, 100 μm. (**G**) Quantification of the percentage of EGFP^+^ cells in different groups. Data are shown as means ± standard derivation. Different letters above the error bar indicate statistical differences (*p* < 0.05). U6, U6 promoter; sgRNA, spacer of sgRNA; scaffold, guide RNA scaffold for the *Streptococcus pyogenes* Cas9; CMV, the promoter of mouse cytomegalovirus; Cas9, zebrafish codon-optimized version of *Streptococcus pyogenes* Cas9; SV40, the simian virus 40 enhancer and early promoter; NeoR, neomycin resistance gene; PuroR, puromycin resistance gene; EGFP, enhanced green fluorescent protein; Ntsf1, sgRNA target sequence of Nile tilapia *sf1*.

**Figure 2 biology-12-00336-f002:**
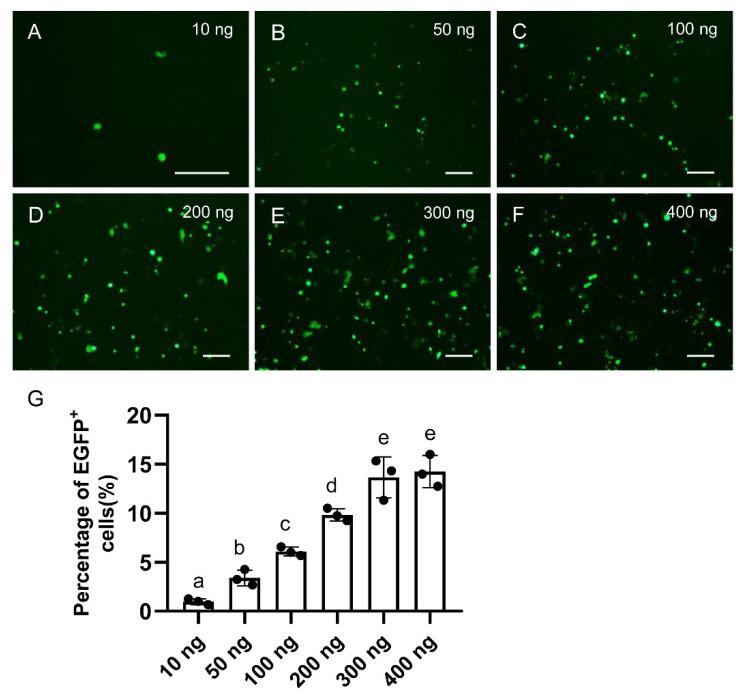
The dosage optimization of pCas9-mU6sgNtsf1 on its gene editing efficacy. (**A**–**F**) SG3 cells in 24-well plates were transfected with pCas9-mU6sgNtsf1 at different doses (10 ng, 50 ng, 100 ng, 200 ng, 300 ng and 400 ng) and pGNtsf1 (200 ng), the cells were monitored under fluorescent microscopy to evaluate the gene editing efficacy 48 h after transfection. Scale bars, 100 μm. (**G**) Quantification of the percentage of EGFP^+^ in different groups. Data are shown as means ± standard derivation. Different letters above the bars represent significant differences between the groups (*p* < 0.05).

**Figure 3 biology-12-00336-f003:**
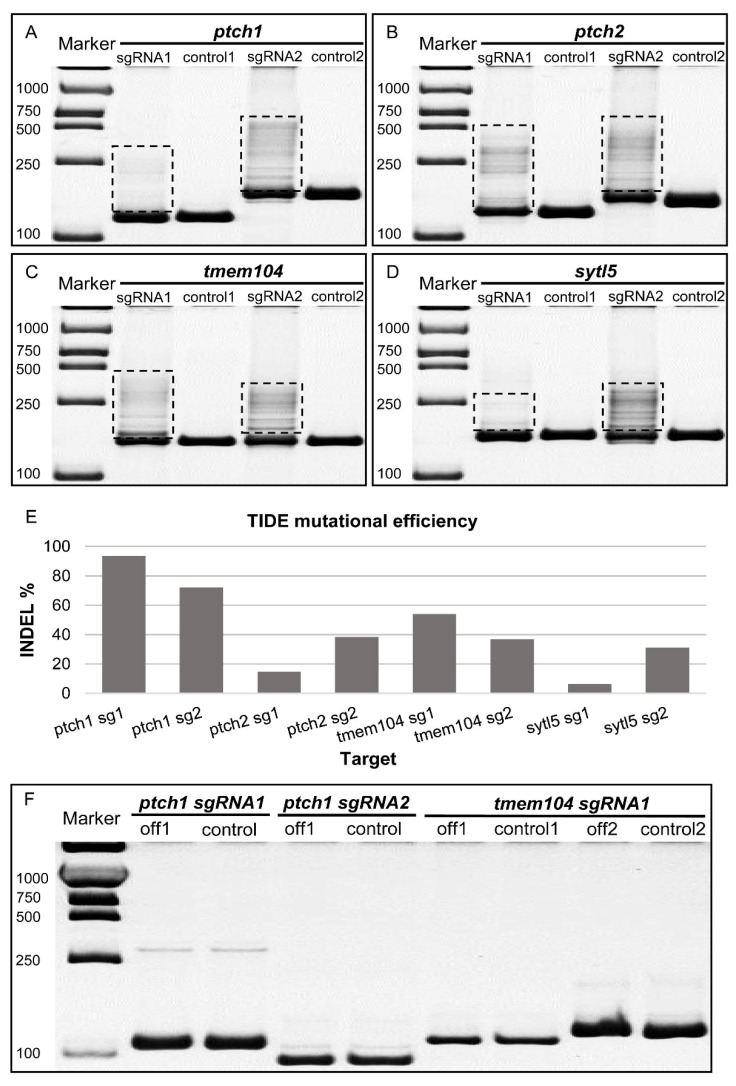
pCas9-mU6sgRNA efficiently edited the endogenous genes *ptch1*, *ptch2*, *tmem104* and *sytl5* in medaka cultured cells. (**A**–**D**) HMA (heteroduplex mobility assay) of DNA templates extracted from SG3 cells transfected with pCas9-mU6sgRNA. pCas9-mU6sgRNA plasmids targeting four endogenous genes *ptch1 (***A**), *ptch2* (**B**), *tmem104* (**C**) and *sytl5* (**D**) were transfected into SG3 cells, respectively. After G418 selection for 7 days, the genomic DNA was extracted for PCR and then the amplicons were separated with PAGE, giving a main band of homoduplex and upper bands consisting of heteroduplex or homoduplex. Control lanes were amplicons of genomic DNA from wild-type SG3 cells. Boxed areas indicate the heteroduplex or homoduplex DNA bands in amplicons from pCas9-mU6sgRNA transfected SG3. (**E**) TIDE (Tracking of Indels by Decomposition) analysis of amplicons from pCas9-mU6sgRNA vector transfected SG3 cells after G418 selection. (**F**) Off-target detection. Off-target sites were predicted by CCTop (CRISPR/Cas9 target online predictor), specific primer pairs were designed and used to amplify corresponding genomic DNA. Amplicons were separated using PAGE. Off (off1 or off2) lanes were amplicons from genomic DNA of pCas9-mU6sgRNA transfected SG3 cells. Control lanes were amplicons of genomic DNA from wild-type SG3 cells. Primer pairs are listed in the Appendix A.

**Figure 4 biology-12-00336-f004:**
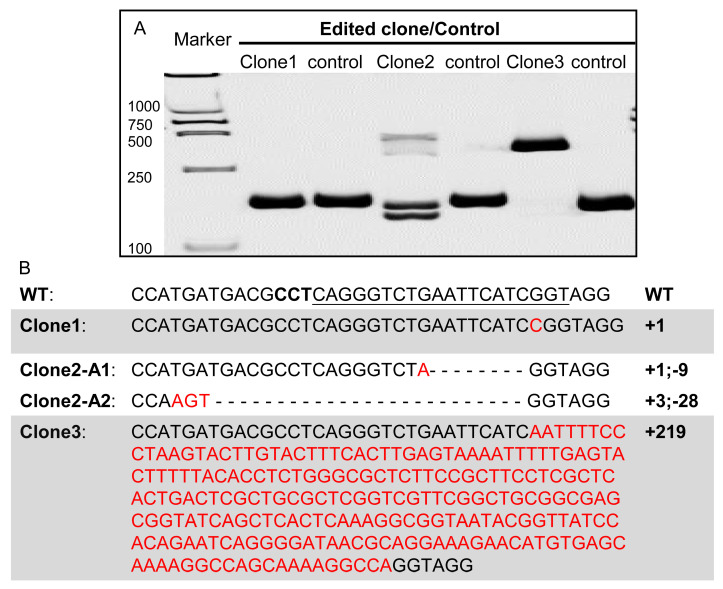
Generation of single cell clones with target gene *sall4* edited. pCas9-mU6sgsall4 was transfected into SG3 cells followed by G418 selection. After enrichment, cells were used to form single-cell clones by limiting dilution. Clones formed by a single cell were picked and genomic DNA was extracted from each clone for PCR amplification. (**A**) Heteroduplex assay of DNA templates extracted from three SG3 single-cell clones with different mutations in *sall4*. Control lanes were amplicons of genomic DNA from wild-type SG3 cells. (**B**) Sequences of *sall4* mutation in single-cell clone 1, 2 and 3. PCR amplicons from DNA templates of clone 1 and clone 3 were directly used for sequencing and PCR amplicon from DNA template of clone 2 was purified and ligated into cloning vector and transformed into *E. coli* for colony picking, five *E. coli* single colonies were selected for plasmid extraction and sequencing. The sgRNA target site is indicated by underline and PAM sequence is indicated in bold. Base insertions are shown in red font, and base deletions are shown in dashes.

**Figure 5 biology-12-00336-f005:**
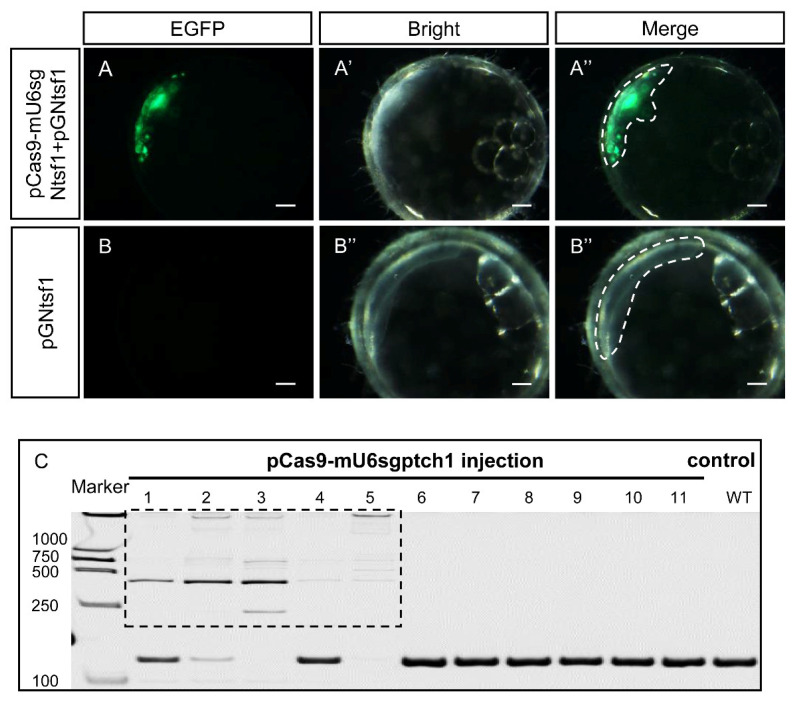
pCas9-mU6sgRNA mediated the exogenous and endogenous gene editing in medaka in vivo by one-cell embryo microinjection. (**A**–**B″**) Microscopy images of medaka embryos injected with pCas9-mU6sgNtsf1 and the reporter vector pGNtsf1 (**A**–**A″**) or injected with pGNtsf1 alone (**B**–**B″**) 24 h after injection. Scale bars, 100 μm. (**C**) HMA of DNA templates extracted from embryos injected with pCas9-mU6sgptch1. Genomic DNA of each embryo was extracted separately. Boxed area indicates the heteroduplex or homoduplex DNA bands in amplicons from pCas9-mU6sgptch1 injected medaka embryos.

**Figure 6 biology-12-00336-f006:**
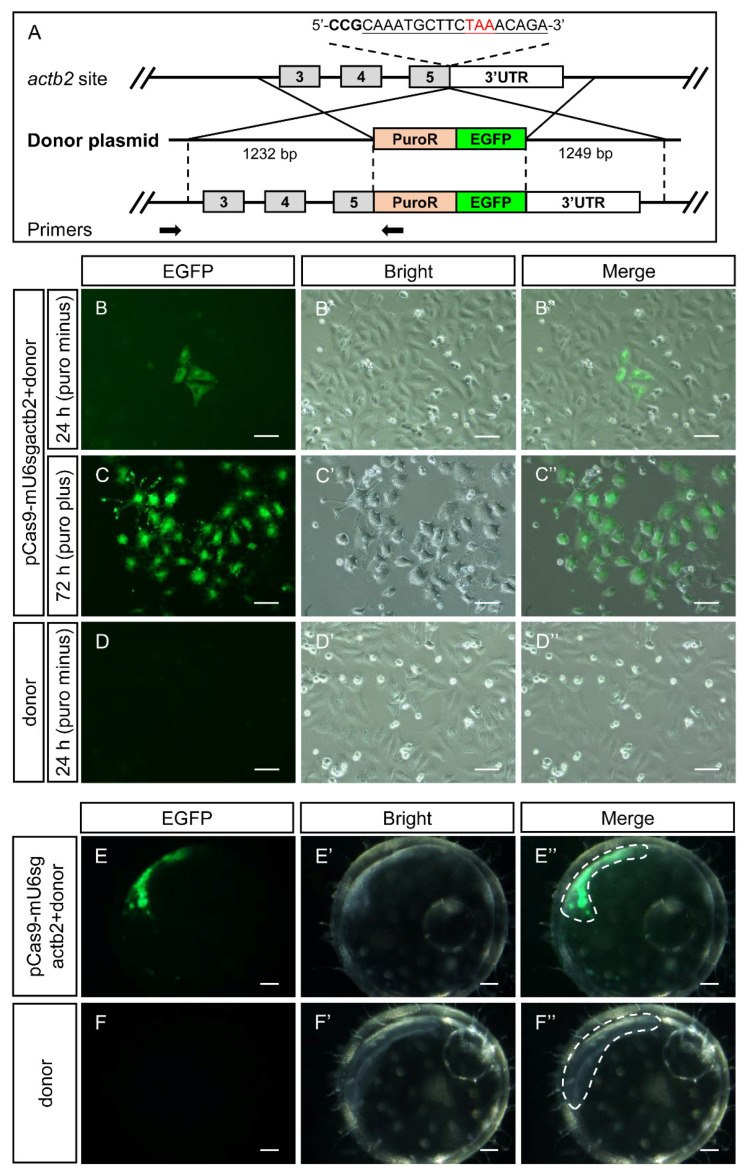
pCas9-mU6sgRNA efficiently mediated the reporter gene puroR-EGFP knock-in in medaka cultured cells and embryos. (**A**) Schematic diagram of gene targeting strategy at medaka *actb2*. The donor plasmid containing a promoter-less puroR-EGFP gene was constructed for targeting *actb2*. Stop codon is shown in red font. The sgRNA target site is indicated by underline and PAM sequence is indicated in bold. PuroR stands for puromycin resistance gene. A pair of primers used for genotyping are indicated by arrows. The length of the left and right homologous arm are 1232-bp and 1249-bp, respectively. (**B**–**C″**) Representative microscopy images of EGFP^+^ SG3 cells before (**B**–**B″**) and after (**C**–**C″**) 1 mg/mL puromycin selection. (**D**–**D″**) Representative microscopy images of the control cells transfected only with the donor plasmids. Scale bars, 50 μm. (**E**–**F″**) Microscopy images of medaka embryos injected with pCas9-mU6sgactb2 and the donor plasmid (**E**–**E″**) or injected with the donor plasmid alone (**F**–**F″**) 24 h after injection. Scale bars, 100 μm.

## Data Availability

All data used to support the findings of this study are included within the article and they are also available from the corresponding author upon request.

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
