# Peer review of "Establishment of an Integrated CRISPR/Cas9 Plasmid System for Simple and Efficient Genome Editing in Medaka In Vitro and In Vivo"

_biology, 2023, doi:10.3390/biology12020336_

Round 1
Reviewer 1 Report
In this manuscript, the authors demonstrate that the genetic relationship of U6 promoter is critical to gene editing efficiency in fish cultured cells, and a simple and efficient CRISPR/Cas9 expression system suitable for medaka gene knock-out and knock-in at a specific site in vitro and in vivo has been established. It provides critical information and an integrated vector-based sgRNA and Cas9 expression system, especially applied in the genome editing of fish cell lines. It's an interesting topic and deserved to be considered to be published in the journal Biology after major revision for the promotion of functional studies of genes in fish cell lines by CRISPR/Cas9 expression system and precision breeding of various fish species in aquaculture. Some points in this manuscript listed below still need to be improved.
1. In Lines 54-58, "Although the CRISPR/Cas9 system has been successfully applied to genome editing in some fish species, such as zebrafish (Danio rerio)[14], medaka (Oryzias latipes)[15], Nile tilapia (Oreochromis Niloticus)[16], southern catfish (Silurus meridionalis Chen)[17], grass carp (Ctenopharyngodon idellus)[18] and so on, its application in fish cultured cells is still challenged and limited[19-25]." The "Chen" behind southern catfish (Silurus meridionalis) should be removed. The fish in Reference #18 is rare minnow (Gobiocypris rarus), which is a fish model for GCRV infection, not grass carp (Ctenopharyngodon idellus). The species name of Nile tilapia "Niloticus" is in a lower-case letter "niloticus". The sentence will be corrected to:
"Although the CRISPR/Cas9 system has been successfully applied to genome editing in some fish species, such as zebrafish (Danio rerio)[14], medaka (Oryzias latipes)[15], Nile tilapia (Oreochromis niloticus)[16], southern catfish (Silurus meridionalis)[17], rare minnow (Gobiocypris rarus)[18] and so on, its application in fish cultured cells is still challenged and limited[19-25]."
2. The long sentence between Lines 109-118 In 2.2 Plasmids of Materials and Methods: "Briefly, take Nile tilapia 109 validated sf1 sgRNA (5’-GGCTGTGTACTGGTACTGGG-3’) [34] insertion in pCas9- 110 zU6sgRNA for example,.............., these three kinds of DNA 117 fragments were assembled together according to manufacturer’s instructions." is hard to understand and should be separated into four sentences as below.
Briefly, take Nile tilapia validated sf1 sgRNA (5’-GGCTGTGTACTGGTACTGGG-3’) [34] insertion in pCas9-zU6sgRNA for example. Primer pairs of zU6MulF (5’-CTTGACGAGTTCTTCTGAACGCGTCTCGAGCCTCTAGA-3’) and zU6sgNtsf1R (5’-CCCAGTACCAGTACACAGCCCGAACCAAGAGCTGGAGGGA-3’) were used to amplify the upstream sequence, and primer pairs of sgNtsf1F (5’-GGCTGTGTACTGGTACTGGGGTTTTAGAGCTAGAAATAGC-3’) and scaffoldSalR (5’-AATTGGCGGTCGACTGGCGTAATAGCCAAC-3’) were used to amplify the downstream sequence. The pCas9-zU6sgRNA plasmid was linearized by restriction enzyme Mlu I and Sal I. These three kinds of DNA fragments were assembled together according to manufacturer’s instructions.
3. In line 396, "grass carp kidney cells JAM-A" is wrong according to Reference #20. JAM-A is the edited target gene Junctional adhesion molecule-A encoding JAM-A protein as GCRV receptor. The grass carp kidney cells are named "CIK" not "JAM-A".
4. In Figure 1. and Figure S2. The effect of U6 promoters derived from different species on gene editing efficacy in medaka cultured cells SG3 and MES1 with pGNtsf1 reporter vector 48 h after transfection. Both two Figures lacked the positive control by using pCVpf, the EGFP reporter vector expressing PuroR-EGFP to show transfection efficiency of SG3 and MES1 cells. As the authors mentioned, due to the limited efficiency of homologous recombination and repair, the actual editing efficiency may be much higher than the efficiency directly observed.
5. As the authors mentioned in the manuscript, by one-cell embryo microinjection, pCas9-mU6sgRNA can effectively mediate gene knock-out in vivo as well. Furthermore, the gene knock-in at a specific site was efficiently achieved in vitro as well as in vivo through the application of pCas9-mU6sgRNA. For in vivo genome editing in medaka by CRISPR/Cas9 expression plasmid, the authors demonstrated EGFP successfully expressed in somatic cells of F0 medaka embryos. As in previous studies, the zygotic genes and foreign transgene injected into one-cell embryos will start to turn on transcription from the mid-blastula transition (MBT) as stage 11 in medaka embryonic development. The earlier onset of the exogenous CMV promoter driving Cas9 expression in pCas9-mU6sgRNA should be by the mid-blastula stage and performed genome editing mainly in somatic cells. How about the real success and efficiency of genome editing in the few primordial germ cells (PGCs) to inherit to F1 generation of medaka by this CRISPR/Cas9 vector system? The efficiency of genome editing in vivo by microinjection of sgRNA and Cas9 mRNA or mixed RNPs into one-cell fertilized eggs of medaka should be much better than pCas9-mU6sgRNA vector-based system.
Reviewer 2 Report
Zhang et al used a plasmid strategy of CRISPR/Cas9 gene editing where the sgRNAs where placed under the control of U6 promoters of three different fish species, namely, medaka, nile and zebrafish. The authors tested the efficiency of the plasmids by co-infecting sgRNA plasmids and an EGFP reporter plasmid in the medaka SG3 cells. A successful editing of the reporter gene resulted in green flourescence, thus validating the pU6sgRNA-Cas9 construct. The system was then used to edit some genes in the medaka fish.
The study is well designed and the experiments were well planned and executed. The manuscript is also properly written.
Major Comments
The findings of the study is quiet relevant to the field of application of CRISPR/Cas gene editing in aquaculture. In particular, the study expands the CRISPR/Cas tool box in this field given that use of plasmid-based CRISPR/Cas editing in fish cell-lines is still plaqued with numerous challenges, some of which are low transfection and editing efficiencies. I´ll recommend that the authors update this aspect of the manuscript and add the suggested references (see attached pdf comments)
However, the authors did not provide justification for the use of plasmid-based strategy over the newer and more efficient RNP strategy. Further, for the cell line aspect of the study, only the SG3 medaka cell line was used and the authors concluded that genetic relationship of U6 promoter is critical to gene editing efficiency in fish cultured cells. The authors did not test the three pU6sgRNA constructs of also in a nile and/or zebra fish cell lines to see whether similar phenomenon will be observed. For example, testing the three pU6sgRNA promoters in, e.g a nile fish cell line to show that ntU6sgRNA construct in the nile cell line had the best (higher/better) GE efficiency compared to the mU6sgRNA and the zU6sgRNA.
Minor Comments: Please see attached pdf.

Reviewer 3 Report
The manuscript “Establishment of an integrated CRISPR/Cas9 plasmid system for simple and efficient genome editing in medaka in vitro and in vivo” introduced a method to do genome editing with CRISPR/Cas9 in medaka. The topic fits the issue. I am not an expert in fish. As a biologist, I think the authors overemphasized the editing efficiency in the manuscript. CRISPR/Cas9 has been successfully applied to genome editing in a lot of species, including some fish species, which limited the novelty of this work. From my point of view, based on the existing data, I do not think this manuscript is a strong candidate.
Concerns:
1. Since U6 promoter shows highest activity in the corresponding species, I do expect to see highest expression of sgRNA in pCas9-mU6sgNtsf1. I am wondering about the significance and reasonableness of comparing the different U6 promoter activities in Figure 1. I think it’s better to put it in supplementary figure.
2. To my understanding, transfection efficiency correlated with the DNA amount in transfection experiments. Higher amounts of DNA show higher transfection efficiency. The authors should add more data to show the dosage effect they saw in Figure 2 is not caused by biases in transfection efficiency.
3. In Figure 3 and Figure 4, the authors enriched the cells with G418 for 7 days after transfection. I think it’s not a proper way to estimate the editing ratio. Long-time G418 treatment will enrich sgRNA and Cas9-containing cells which show high expression of both sgRNA and Cas9. Consistent expression of sgRNA and Cas9 in the cell leads to a higher editing ratio. So, the experimental pipeline enables us to see only edited cells and finally see elevated editing ratio. The actual editing ratio is much lower than what they claimed in the paper.
4. The in vivo knock-in experiment is interesting. It would be much better if the author could validate the protein expression by western blot to make sure the GFP-fused protein is properly expressed.
Round 2
Reviewer 1 Report
The authors revised the manuscript according to the comments and suggestions of the reviewer. In point 5 of comments and suggestions as the reviewer mentioned "The efficiency of genome editing in vivo by microinjection of sgRNA and Cas9 mRNA or mixed RNPs into one-cell fertilized eggs of medaka should be much better than pCas9-mU6sgRNA vector-based system", the author made a reasonable response as below. Please include these sentences in the Discussion to convince us this pCas9-U6sgRNA vector-based system is also useful for the in vivo application.
Response 5: This's a good question. As the reviewer’ say, maybe it is better to gain genome editing offspring by microinjection of sgRNA and Cas9 mRNA or mixed RNPs into one-cell fertilized eggs of medaka. However, this pCas9-mU6sgRNA vector-based system might provide an alternative choice for the study of genes with lethal effects at the early stage of embryonic development, and the delayed gene editing by this plasmid-based system might be more conducive to study their functions in the later development stage.
Author Response
Thanks a lot for the suggestion of the reviewer. At the end of the first paragragh in the Discussion, we have added “Meanwhile, pCas9-mU6sgRNA can efficiently mediate gene knock-our ot knock-in at a specific site in embryos as well. Although it may be better to gain genome editing offsprings by microinjection of sgRNA and Cas9 mRNA or mixed RNPs into one-cell embryos, the pCas9-mU6sgRNA provides an alternative choice for the study of genes with lethal effects at the early stage of embryonic development. Because the expression of sgRNA and Cas9 in pCas9-mU6sgRNA may be initiated until blastula, the delayed gene editing might be conducive to study the functions of these lethal genes in embryonic development at the late stage.”
Reviewer 3 Report
Frankly, this manuscript fits the topic of the journal and the issue. I expected the authors to add some new data to validate their results in the revised manuscript to make it better. Based on the revised manuscripts I got, I do not think the authors directly addressed the reviewers’ concerns by providing experimental evidence. But I still think the manuscript greatly improved in terms of expression or statement which makes it more precise and clearer. It might be interesting to researchers who work on fish, especially medaka, even though it’s not attractive to me, personality. I noticed there is a new author added in the revised manuscripts. Since nothing new added to the revised manuscript, it’s not common. I think the authors need to explain her/his contributions in the revised manuscript.
Author Response
We are sorry that we forgot to add the author Jie Wang in our first draft due to negligence. In fact, Jie Wang has graduated as a graduate student. She was the first graduate student in our research group to carry out gene editing in fish cultured cells. She mainly used sgRNA and Cas9 mRNA or mixed RNPs and the gene editing effiency was very low. She had constructed multiple plasmids, isolated U6 promoters and so on. After her graduation, this job was transferred to the next graduate student, Zeming Zhang. During our revision process, we realized that she had made a lot contributions to this job in terms of conceptualization, methodology and investigation. So we added her name in the revised draft.